# Fine-Tuned Ecological Niche Models Unveil Climatic Suitability and Association with Vegetation Groups for Selected *Chaetocnema* Species in South Africa (Coleoptera: Chrysomelidae)

Francesco Cerasoli *[ID], Paola D'Alessandro and Maurizio Biondi

Department of Life, Health and Environmental Sciences, University of L'Aquila, Via Vetoio Coppito, 67100 L'Aquila, Italy; paola.dalessandro@univaq.it (P.D.); maurizio.biondi@univaq.it (M.B.)
* Correspondence: francesco.cerasoli@univaq.it

**Abstract:** Despite beetles (Coleoptera) representing most existing animal species, the ecological and biogeographical factors shaping their distribution are still unclear in many regions. We implemented state-of-the-art ecological niche models (ENMs) and niche overlap analysis to investigate climate–occurrence patterns for five flea beetle species of the genus *Chaetocnema* in South Africa (*C. brincki*, *C. danielssoni*, *C. darwini*, *C. gahani,* and *C. natalensis*). ENMs were fitted through Maxent and Random Forests, testing various parameterizations. For each species, tuned ENMs attaining good discrimination on spatially independent test data were selected to predict suitability across the study region and individuate its main climatic drivers. Percentage coverage of climatically suitable areas by seventeen Afrotropical vegetation formations was also computed. Predicted suitable areas do not extend far away from known presence localities, except for *C. brincki* and *C. gahani* in north-eastern South Africa. Temperate grasslands and shrublands cover most of suitable areas for *C. brincki* and *C. gahani*, along with warm temperate forests, as well as for *C. danielssoni*, in this case being followed by tropical flooded and swamp forests. Climatic suitability for *C. darwini* mainly relates to the Mediterranean grasslands and scrublands of the southern coastal region, while suitable areas for *C. natalensis* encompass various vegetation formations, coherently with its wide distribution. The environmental niche of *C. danielssoni* significantly overlaps with those of the wide-ranging *C. darwini* and *C. natalensis*, suggesting that historical factors, rather than low climatic tolerance, has determined its restricted distribution in the Western Cape Province. Maxent and Random Forests were confirmed to be of great help in disentangling the environment–occurrence relationships and in predicting suitability for the target species outside their known range, but they need to be properly tuned to perform at their best.

**Keywords:** flea beetles; ecological niche models; biogeography; Maxent; Random Forests; *Chaetocnema*; South Africa; niche conservatism

## 1. Introduction

Insects are by far the most diversified animal class, with more than one million species already described and at least 5 million ones estimated to currently exist [1]. Their astonishing morphological and functional diversity translates into fundamental ecological roles, spanning from plant pollination to nutrient recycling within different environmental matrices (e.g., soil, freshwaters), to food supply for secondary consumers and pests' bio-control [2–4]. Despite their invaluable importance for the maintenance of the ecological processes shaping the Earth's biodiversity, large knowledge gaps still affect the taxonomy (i.e., Linnean shortfall) and biogeography (i.e., Wallacean shortfall) of several insect taxa [5]. Among insects, beetles (order Coleoptera) represent a hyper-diversified group comprising about 25% of all the animal species described up to date [4]. As for the majority of insect

groups, beetle species richness is particularly high in tropical and subtropical regions of Africa, Asia and South America, most of which have not been extensively sampled yet [5]. Thus, lack of comprehensive occurrence data in these regions is likely limiting our understanding of the ecological drivers constraining the current distribution of several beetle species, which in turn is detrimental not only to the task of limiting the above-mentioned Wallacean shortfall but also to the design of effective conservation measures. Nonetheless, even relatively sparse occurrence data may provide valuable information about the ecological requirements of the target species [6], as well as about macroecological patterns [5]. For instance, when presence records are accompanied by geographical coordinates, abiotic conditions at occurrence localities can be extracted from the continuously growing set of online databases storing environmental gridded surfaces at various spatial and temporal resolutions, such as Worldclim [7] or CHELSA [8], and then be contrasted to conditions characterizing the study region through through ecological niche modeling techniques [9]. In this manner, one can assess the environment-occurrence relationships affecting the observed distribution of the target species, and the fitted model can successively be projected in the geographic space to estimate the species' potential distribution [10]. Indeed, species' distribution at regional-to-global scales is primarily affected by abiotic factors such as climate and topography [11,12], and insects make no exception to this general rule [13]. The implementation of ecological niche models (ENMs) in insect-related studies has gained increasing popularity in the last decade, ranging from global assessments of the effects of different future warming scenarios upon insect diversity [14], to the regional quantification of potential climate-related range shifts in mountainous areas for selected insect species [15–17], to the estimation of invasive potential for exotic species [18,19] with particular regard to those representing pests of plants being of agricultural importance [20–22]. In this contribution, we implemented a state-of-the-art ecological niche modeling framework to investigate climate-related drivers influencing the distribution of five flea beetle species of the genus *Chaetocnema* Stephens (Chrysomelidae: Galerucinae: Alticini) in southernmost Africa. *Chaetocnema* is a worldwide distributed genus of phytophagous beetles occurring in the Afrotropical region with more than 100 species [23]. These flea beetles are primarily found within moist habitats and grasslands, associated with various plant species belonging to the Amaranthaceae, Polygonaceae, Cyperaceae and Poaceae families [24], and taxonomic knowledge for them is still accumulating [25]. The species selected for our analyses are endemic or sub-endemic to the Republic of South Africa, Lesotho and Swaziland. The region encompassed by these countries hosts three biodiversity hotspots (i.e., Cape Floristic Region, Succulent Karoo, and Maputaland-Pondoland-Albany), and its beetle fauna is being increasingly investigated in terms of environmental preferences and related distributional patterns [26–29]. Here, starting from previous studies about *Chaetocnema* species in the Afrotropics [23,24], we move forward by identifying the climatic drivers behind the observed distribution of the selected species, by modeling the potential distribution of these species in the study region so as to also inform future sampling campaigns, and by highlighting inter-specific divergences in climatic suitability and in the associations between the latter and the vegetation formations [30] characterizing the study region.

## 2. Materials and Methods

### 2.1. Target Species and Study Area

The target *Chaetocnema* species were *C. brincki* (Bechyné, 1959), *C. danielssoni* Biondi & D'Alessandro, 2006, *C. darwini* Bryant, 1928, *C. gahani* Jacoby, 1897 and *C. natalensis* Baly, 1877 (Figure 1). In terms of chorotypes (i.e., biogeographical units defined upon groups of non-randomly overlapping species' distributions [31]), *C. brincki* belongs to the South-East African chorotype, *C. darwini* and *C. natalensis* share the Southern African chorotype, *C. danielssoni* is a South-West Afrotropical species, while *C. gahani* falls into the Eastern Afrotropical chorotype [24,32]. Based on available auto-ecological information, these species apparently share a preference for wet habitats.

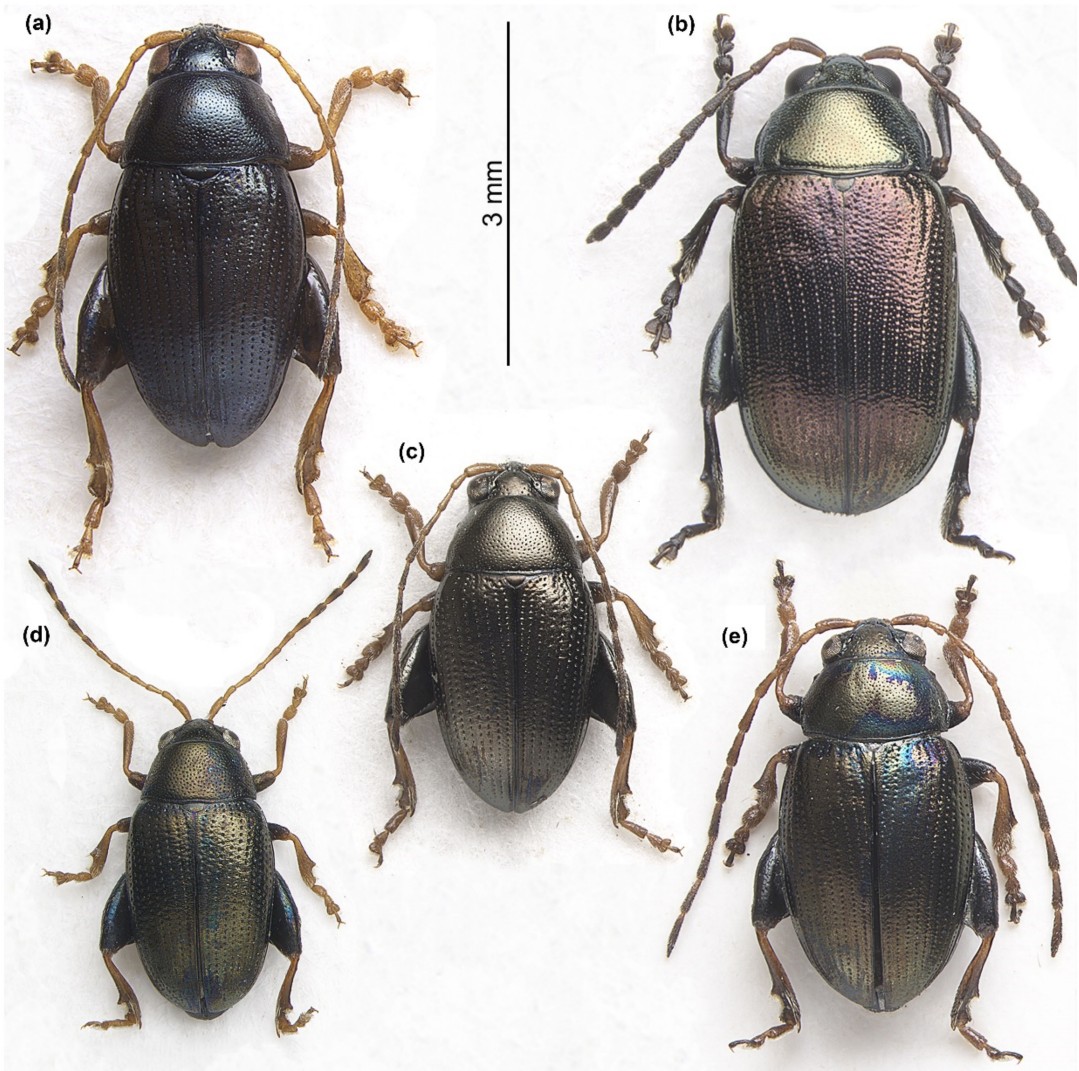

**Figure 1.** Habitus of the target flea beetle species of the genus *Chaetocnema* Stephens, whose climate—occurrence relationships and associations with vegetation groups in South Africa were analyzed; (**a**) *Chaetocnema gahani* Jacoby, 1897; (**b**) *C. natalensis* Baly, 1877; (**c**) *C. darwini* Bryant, 1928; (**d**) *C. danielssoni* Biondi & D'Alessandro, 2006; (**e**) *C. brincki* (Bechyné, 1959).

Occurrence records of the target species in WGS84 geographic coordinates system were retrieved from Biondi & D'Alessandro (2006) [33] and unpublished data (MB), mostly collected between 1980 and 2005. The data gathering led to 6 occurrence records for *C. brincki*, 10 for *C. danielssoni*, 37 for *C. darwini*, 12 for *C. gahani* and 26 for *C. natalensis*; the corresponding coordinates are provided in Supplementary Materials Table S1.

The study region (spatial extent: Longitude 16.44°E–38.00°E, Latitude 22.12°S–36.00°S) corresponds to the administrative borders of the Republic of South Africa (excluding Prince Edward Islands), Lesotho and Swaziland, and is characterized by a noticeable variety of vegetation formations (Figure 2, Table 1). In order to provide a graphical representation of the extent of occurrence (EOO) of each target species within the study region, $\alpha$-hull polygons were drawn through the 'alphahull' [34] R [35] package, setting $\alpha$ = 2.5. An $\alpha$-hull derives from a Delaunay triangulation over the considered set of points, and the $\alpha$ parameter determines the shape and number of polygons forming the $\alpha$-hull because all the lines within the Delaunay triangulation exceeding the average line length by $\alpha$ times are removed from the hull [36]. The $\alpha$ value we chose is the average between $\alpha$ = 2

recommended by the International Union for Conservation of Nature (IUCN) [36] for some species and $\alpha = 3$ recommended by Burgman and Fox [37].

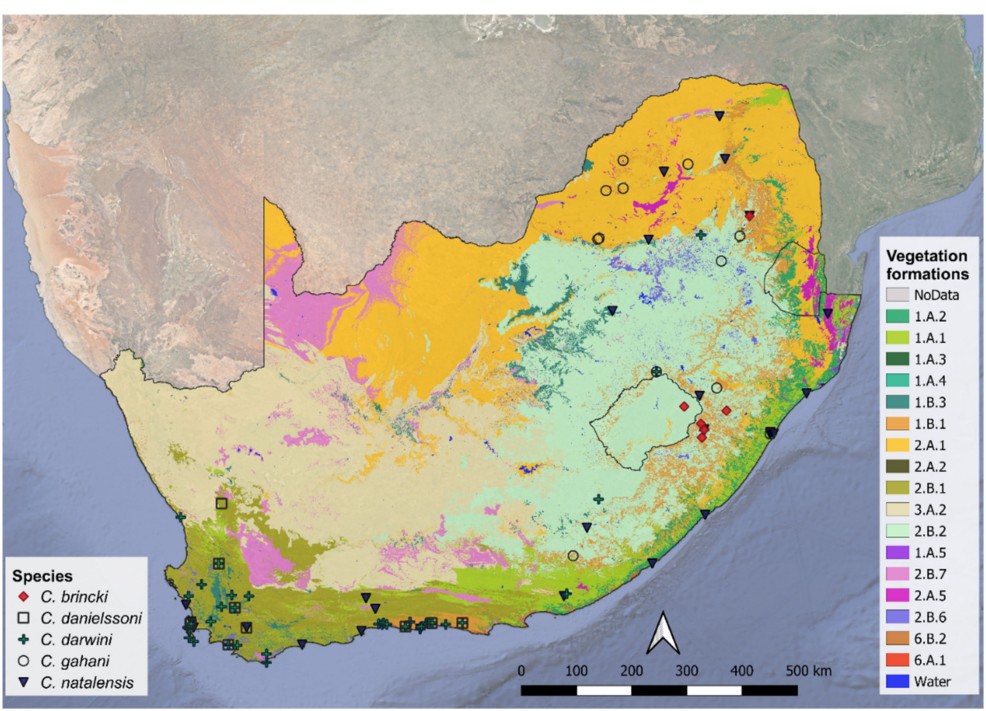

**Figure 2.** Map of the study region (WGS84 geographic coordinate system); occurrence records of the target *Chaetocnema* species are overlaid on a raster map (90 m cell resolution) representing vegetation formations of the Republic of South Africa, Lesotho and Swaziland. The name of the vegetation formation corresponding to each of the alphanumerical codes in the legend are reported in Table 1.

**Table 1.** Vegetation formations of Africa and corresponding alphanumeric codes.

| Formation Name | Formation Code |
|---|---|
| Tropical Seasonally Dry Forest | 1.A.1 |
| Tropical Lowland Humid Forest | 1.A.2 |
| Tropical Montane Humid Forest | 1.A.3 |
| Tropical Flooded & Swamp Forest | 1.A.4 |
| Mangrove | 1.A.5 |
| Warm Temperate Forest | 1.B.1 |
| Temperate Flooded & Swamp Forest | 1.B.3 |
| Tropical Lowland Grassland, Savanna & Shrubland | 2.A.1 |
| Tropical Montane Grassland & Shrubland | 2.A.2 |
| Tropical Freshwater Marsh, Wet Meadow & Shrubland | 2.A.5 |
| Mediterranean Scrub & Grassland | 2.B.1 |
| Temperate Grassland, Meadow & Shrubland | 2.B.2 |
| Temperate & Boreal Freshwater Marsh, Wet Meadow & Shrubland | 2.B.6 |
| Salt Marsh | 2.B.7 |
| Warm Desert & Semi-Desert Scrub & Grassland | 3.A.2 |
| Tropical Cliff, Scree & Other Rock Vegetation | 6.A.1 |
| Temperate & Boreal Cliff, Scree & Other Rock Vegetation | 6.B.2 |

### 2.2. Bioclimatic Variables and Vegetation Formations

Worldwide gridded climate surfaces representing 19 temperature- and precipitation-related "bioclimatic" variables were downloaded as raster files from the Worldclim 2.1 online repository [7] (https://worldclim.org/data/worldclim21.html, accessed on 7 January 2022) in WGS84 geographic coordinate system at 2.5 arc-minutes (~5 km at the equator) cell resolution. Spatial information about vegetation formations (Table 1) characterizing the study region was instead retrieved from a raster map of terrestrial ecosystems of Africa [30] providing a hierarchical classification (i.e., class, subclass, formation, division, macro-group) of African vegetation at 90 m cell resolution.

Subsequent spatial analyses were performed by means of the 'raster' [38] and 'terra' [39] R packages.

First, each raster was cropped to the extent of the study region and masked to its borders. Secondly, to avoid geographical biases in the selection of the background points needed to fit the ecological niche models (see *Model Fitting, Validation and Projection*), the cropped raster maps were reprojected to the equal-area projection coordinate system 'Africa sinusoidal' (EPSG code: 102011). Finally, to adjust the spatial information about vegetation formations to the same resolution of the bioclimatic variables, we used the 'aggregate' function from the 'raster' package to create, for each of the 17 vegetation formations, a new raster file reporting the percentage coverage by that formation within each aggregated raster cell (the aggregation factor corresponded to the ratio between the resolution of the reprojected bioclimatic variables and that of the reprojected 90 m resolution raster of vegetation formations).

### 2.3. Model Fitting, Validation and Projection

To avoid feeding the ecological niche models (ENMs) with several predictors affected by multicollinearity, which would lead to biased estimates of the relative importance of the single predictors [40], we performed a variance inflation factor (VIF) analysis on the full set of 19 bioclimatic variables through the 'usdm' [41] R package: the variables exceeding the conservative VIF = 5 [11] threshold across the study region were discarded.

We selected two machine learning algorithms, Random Forests [42] and Maxent [43], to fit the ENMs. Random Forests (hereafter RF) works by fitting hundreds of classification or regression trees (based on the type of response variable, namely categorical versus continuous), each time using a bootstrapped sample of the training data and a random sample of the available predictors [44]. The proportion of training data used to fit each single tree, the number of predictors selected at each iteration, the 'depth' (i.e., number of node splits) of the trees and the minimum number of observations belonging to a terminal node (i.e., a 'leaf' of the tree) can all be controlled within most R packages permitting the implementation of RF, and all contribute to shape the final model [44]. Differently, Maxent relies on the maximum entropy principle: once values of the predictors are sampled from the species' occurrence localities and from a set of background points large enough to represent as comprehensively as possible the environmental conditions characterizing the study region, the algorithm uses a combination of various feature classes (i.e., transformations of the single predictors) to estimate the maximum entropy distribution (i.e., the one closest to the uniform distribution but constrained by the means of the considered features across occurrence localities) of the suitability/probability of occurrence for the target species across the study region [43,45].

RF and Maxent have been extensively implemented in ecological niche modeling in the last fifteen years [46–49] and they often ranked among the top-performing presence-background models in terms of predictive accuracy within studies comparing multiple algorithms on a same dataset [50–52]. Nonetheless, in many previous studies using ENMs for macroecology-, biogeography- or conservation-related inferences, Maxent and RF were implemented relying on their default parameterization, which may not be appropriate to the specific dataset and modeling task, and thus may lead to less reliable predictions [44,53,54]. Here, we fine-tuned both algorithms choosing the parameterization setting(s) leading to

the highest accuracy on withheld data. Maxent tuning was accomplished by means of the 'ENMeval' R package [55], testing various combinations of feature classes and values of the regularization multiplier (Table 2), the latter controlling the degree of smoothing in the model via the shrinking of features' coefficients [45]. Differently, to tune the RF models, fitted using the 'randomForest' R package [56], we tested different combinations of the number of trees composing the final model and of the size of terminal nodes (Table 2).

**Table 2.** Cross-validation strategy and tuning parameters used to fit and validate the ENMs for each target *Chaetocnema* species.

| Species | CV Strategy | Maxent Tuning | RF Tuning |
|---|---|---|---|
| *C. brincki*<br>*C. danielssoni*<br>*C. gahani* | buffered LOO | FC: linear, quadratic, hinge<br>RM:1, 1.5, 2, 2.5, 3 | Number of trees: 500, 1000, 2000<br>Size of terminal nodes: 1, 5,<br>$0.25 \times n$ training occur. |
| *C. darwini*<br>*C. natalensis* | Checkerboard blocking | FC: linear, quadratic, hinge,<br>product, threshold<br>RM:1, 1.5, 2, 2.5, 3 | Number of trees: 500, 1000, 2000<br>Size of terminal nodes: 1, 5,<br>$0.25 \times n$ training occur. |

CV = cross-validation; LOO = 'leave-one-out'; FC = feature class; RM = regularization multiplier; $0.25 \times n$ training occurr. = one-fourth of the number of occurrences used to train the model.

Along with parameters' tuning, appropriate evaluation of ENMs' predictive performance is fundamental to select the model(s) with the best trade-off between fit on calibration data and generalizability [50,57]. For instance, when spatial autocorrelation (hereafter SAC) structures that are intrinsic to the occurrence data are ignored and ENMs are evaluated on randomly selected subsets of these data, estimates of model accuracy tend to be inflated [57,58]. To avoid this risk, we implemented spatially blocked cross-validation strategies [57,59]. For *Chaetocnema brincki*, *C. danielssoni* and *C. gahani*, which were represented by few occurrences ($n < 15$), we implemented geographically buffered leave-one-out cross-validation (hereafter LOO CV). In LOO CV, a single presence point is iteratively withheld for model testing along with the background points falling within a certain radius around it (we chose a 50 km radius corresponding to about 10 grid cells, see [57]), and the remaining occurrences and background points are used to fit the ENM: in this manner, $n$ models, each calibrated on $n-1$ occurrences, are fitted for the target species. Differently, for *C. darwini* and *C. natalensis* we implemented the checkerboard blocking cross-validation (hereafter Checkerboard CV) strategy. In Checkerboard CV, the available data are grouped in two folds, each composed of non-contiguous blocks encompassing a certain portion of the available presence and background points; the ENMs are then fitted iteratively on each of the two folds and validated on the other one [57]. Checkerboard CV could be implemented for these two species as their higher sample sizes permitted to split the available presences between two folds without reducing too much the number of occurrences on which the single ENMs were calibrated. To design the blocks for Checkerboard CV, we first had to estimate the SAC range in model residuals for *C. darwini* and *C. natalensis*, because blocks should be larger than this range to ensure the spatial independence between training and test data [57]. Thus, for these two species, we fitted 10 preliminary Maxent models and 10 preliminary classification RF models maintaining the default parameters defined for these algorithms in the 'dismo' [60] and 'randomForest' [56] R packages, respectively. Each preliminary model was fitted using all the available occurrences, along with: for Maxent, a random sample of 10,000 background points drawn within a buffer polygon of a 600 km radius around the presence points; for RF, a random sample of 100 background points (see [61] for the lower number of background points chosen for RF) were drawn within a buffer polygon ranging from 10 to 600 km around the presence points. This geographically buffered sampling approach for background points was chosen because it has been shown to enhance the predictive accuracy for both Maxent and RF [61,62]. Moreover, the use of an internal buffer of 10 km for RF aimed at limiting the so-called 'class-overlap' (i.e., presences

and background points sharing the same environmental conditions), which has recently been shown to favor poor predictive performance in RF [44].

Residuals from predictions of these preliminary models on the training data themselves were then used to derive, through the 'ncf' R package [63], correlograms showing how residuals-based Moran's index (I) varied at increasing inter-point distance; indeed, the distance after which Moran's I, computed from model residuals, approaches 0 corresponds to the SAC range [57]. Once we found the optimal block size for *C. darwini* and *C. natalensis*, the corresponding Checkerboard CV structure was defined using the 'spatialBlock' function from the 'blockCV' R package [59], setting the minimum number of calibration points (i.e., presences + background) within each fold to 50. For the remaining species, the LOO CV structure was defined using the 'blockCV' package as well, through the 'buffering' function.

Then, new Maxent and RF models were fitted for each train-test split resulting from the species-specific CV structure, using all the parameter combinations shown in Table 2. Maxent models were fitted through the 'ENMevaluate' function of the 'ENMeval' package, which permits us to evaluate different model parameterizations and select the one matching the user-defined performance criteria [55]; classification RF models were fitted again using the 'randomForest' package, but this time we implemented the "down-sampling" approach (i.e., taking an equal number of presence and background points to fit each individual tree), which has been shown to greatly outperform the 'basic' classification RF [44]. The best performing ENM for each train-test CV split was chosen according to two hierarchical criteria: (i) AUC (i.e., area under the curve of the receiver operating Characteristic plot) computed on the test fold (hereafter AUCtest) $\geq 0.8$, corresponding to good-to-excellent model discrimination [64]; (ii) in case two or more ENMs matched the former criterium, the one with the lowest difference (hereafter AUCdiff) between the AUCtest and AUC computed on the training fold was chosen, so as to penalize the models showing overfitting [54]. In case none of the ENMs fitted for a train-test split attained AUCtest $\geq 0.8$, we chose the model with the highest AUCtest and lowest AUCdiff.

Once the best performing ENM for each train-test split was found, it was projected across the study region, and standardized importance (hereafter Std_Imp) scores of the bioclimatic variables selected for model fitting were computed. Std_Imp scores for the Maxent models were computed through the 'varImportance' function of the 'fitMaxnet' R package [65], while those for the down-sampled RF models were retrieved by means of the internal 'randomForest' function looking at the decrease in the Gini coefficient after randomizing predictors values among the observations [42].

Subsequently, for each species × algorithm combination, we computed the weighted average (hereafter Wgt_Avg) [66] and the weighted standard deviation (hereafter Wgt_StdDev) [67] among the different train-test splits of: (i) predicted suitability across the study region; (ii) Std_Imp scores of the predictors. In both cases, the weight applied to the tuned ENM selected for each split corresponded to the respective AUCtest value. Moreover, partial response curves from the tuned Maxent and RF models fitted for each train-test split were drawn through the evaluation strip approach [68] for the three variables attaining the highest Wgt_Avg Std_Imp scores. In this manner, we could evaluate inter- and intra-algorithm variations in the estimated climate–occurrence relationships and in the derived predictions of climatic suitability.

### 2.4. Associations between Climatic Suitability and Vegetation Formations

Weighted average predictions of climatic suitability obtained for each target species were binarized (i.e., suitable–unsuitable) according to two different thresholds: (i) for the 10th percentile of the Wgt_Avg suitability values sampled on the presence points, a binarization threshold is frequently used when dealing with datasets composed of records gathered over long time periods and by various collectors [69,70]; (ii) Wgt_Avg suitability ≥ 0.8, so that only the pixels showing high predicted suitability across the different ENMs would be considered climatically suitable for the species. Finally, the percentage cover of the 17 vegetation formations was sampled from the pixels predicted as suitable according to each of the two binarization thresholds, in order to assess if areas being climatically suitable for the target species are characterized by prevalent coverage of one or more vegetation formations.

### 2.5. Niche Overlap

To further highlight inter-specific affinities or differences in environmental preferences, we implemented the PCA-Env approach described in Broennimann et al. (2012) [71], by means of the 'ecospat' R package [72]. Specifically, a principal component analysis (PCA) was performed to summarize environmental heterogeneity across the study region through a set of uncorrelated axes derived by extracting values of the 19 bioclimatic variables and the percentage coverage from the 17 vegetation formations corresponding to the target species' occurrence localities and of 10,000 randomly sampled background points. Then, kernel-based density of occurrence (hereafter niche density) in the D environmental space, encompassed by the top two (in terms of explained variance) principal components (hereafter PrinComp), was estimated for each species, and pairwise overlap in niche density was measured for all the 10 species' pairs using the Schoener's *D* metric [73]. Finally, the niche divergence and niche conservatism hypotheses were tested, for each species' pair, through the niche similarity test [71]: in each of the 100 iterations, a simulated niche density was estimated for each species from points randomly sampled within a 120 km-wide buffer around its occurrence points (i.e., to represent the 'available environment'), and the overlap between the simulated densities was computed through the *D* metric; then, the observed *D* value, previously computed for the actual niche densities of the considered species' pair, was compared to the distribution of the 100 simulated *D* values to assess if the former was higher than the 95th percentile (niche conservatism) or lower than the 5th percentile (niche divergence) of the latter.

### 3. Results

The alphahull-based extents of occurrence (EOO) estimated for *Chaetocnema brincki* (Figure 3a) and *C. gahani* (Figure 3d) span eastern Lesotho and the South African region, located between the latter and Swaziland, which is partially included in the Maputaland-Pondoland-Albany biodiversity hotspot [24], with the range of the former species entirely nested within that of the latter. On the other side, the estimated EOO for *C. natalensis* (Figure 3e) spans the entire southern coastal region of South Africa and then extends across the north-east, thus encompassing the EOOs of both *C. brincki* and *C. gahani*. Differently, the EOO of *C. danielssoni* (Figure 3b) is limited to the south-westernmost portion of South Africa, across the Cape Floristic Region and Succulent Karoo hotspots, while the EOO of *C. darwini* (Figure 3c) broadly corresponds to that of *C. danielssoni* in the south-west but then extends to the north-east across Lesotho, partially overlapping with the EOOs of *C. natalensis*, *C. gahani,* and *C. brincki*.

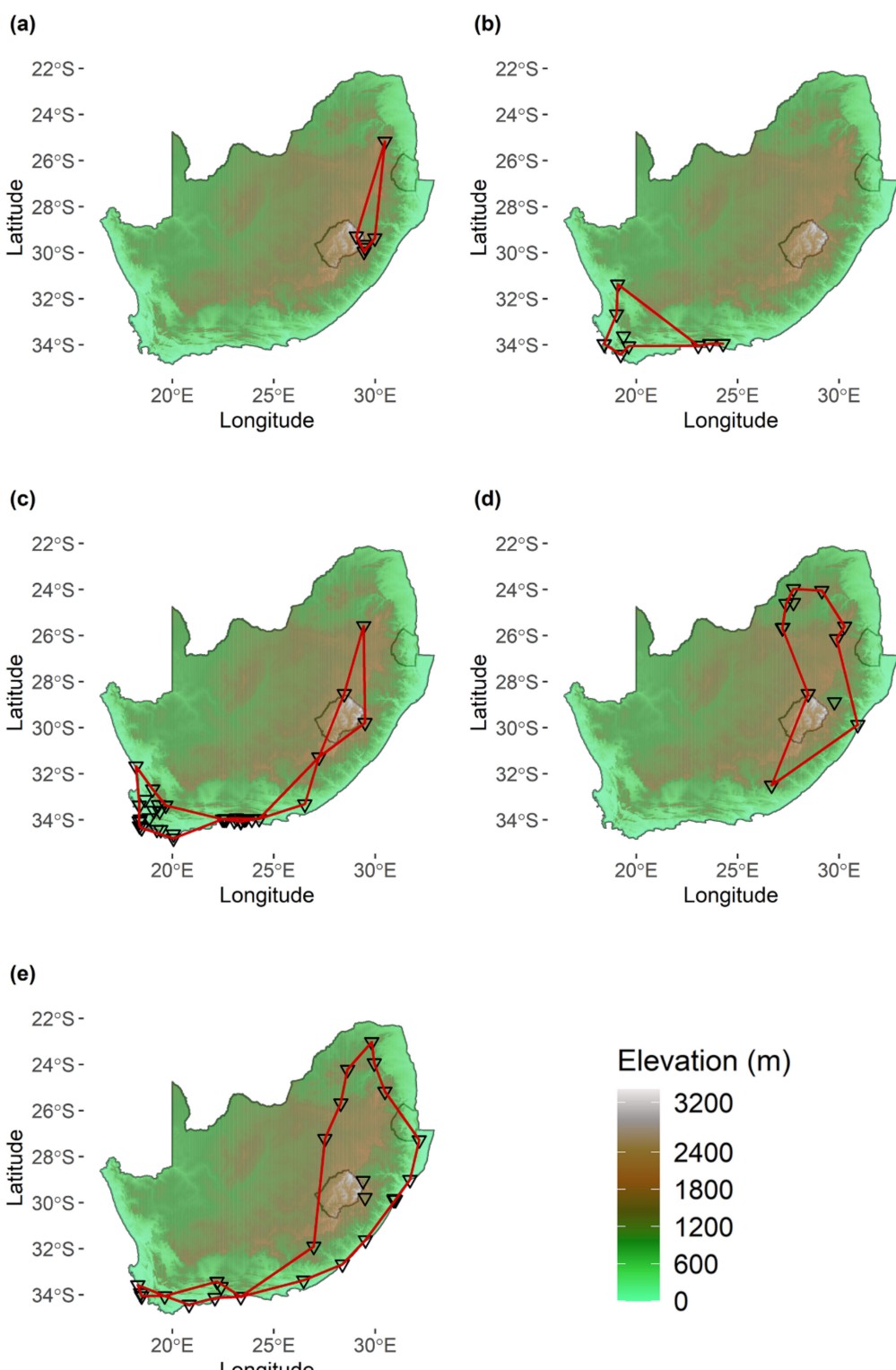

**Figure 3.** Alphahull-based estimated range (red lines) and occurrence records (reversed black-contoured triangles), overlaid on a DEM map showing altitude values within the study region at 30 arc-seconds (i.e., ~1 km²) cell resolution, for: (**a**) *Chaetocnema brincki*; (**b**) *C. danielssoni*; (**c**) *C. darwini*; (**d**) *C. gahani*, and (**e**) *C. natalensis*.

The bioclimatic variables selected for model fitting after the VIF analysis were: bio2 (mean diurnal temperature range), bio3 (isothermality), bio8 (mean temperature of the wettest quarter), bio9 (mean temperature of the driest quarter), bio13 (precipitation of the wettest month), bio15 (precipitation seasonality), and bio19 (precipitation of the coldest quarter).

Within the preliminary Maxent and RF models fitted for *C. darwini* and *C. natalensis*, spatial autocorrelation of model residuals became negligible at an inter-point distance of about 200 km (Supplementary Materials Figure S1), so we chose 250 km as block size for both species. An example of the resulting checkerboard structure is reported in Supplementary Materials Figure S2.

The ENMs' tuning process led to some noticeable differences among the target species in the selected model parameters (Supplementary Materials Table S2), for both Maxent and RF. For instance, while the optimized Maxent models for *C. natalensis* used only linear features with relatively strong smoothing (i.e., regularization multiplier = 3), accurately modeling the climate-occurrence relationships for *C. gahani* required a combination of linear, quadratic and hinge features in 11 out of 12 train-test LOO splits, with the regularization multiplier varying between 1 and 2.5. Considering RF, at least 1000 trees were necessary to get accurate predictions on test data in most of the train-test CV splits for *C. brincki*, *C. darwini* and *C. gahani*, while 500 trees were sufficient for both the Checkerboard CV splits of *C. natalensis* and for half of the LOO splits of *C. danielssoni*. Differently, the size of the terminal nodes in the optimized RF models was equal to five for both the Checkerboard CV splits of *C. darwini* and *C. natalensis*, while it noticeably varied among the train-test splits for the remaining species.

Considering the discrimination performance of the tuned ENMs, Maxent models generally attained higher AUCtest values than RF for all the target species (Supplementary Materials Table S2), with mean ± sd values across the train-test CV splits corresponding to: (a) *C. brincki*: Maxent = 0.94 ± 0.05; RF = 0.79 ± 0.10; (b) *C. danielssoni*: Maxent = 0.95 ± 0.02; RF = 0.88 ± 0.12; (c) *C. darwini*: Maxent = 0.91 ± 0.01; RF = 0.92 ± 0.07; (d) *C. gahani*: Maxent = 0.82 ± 0.11; RF = 0.82 ± 0.16; (e) *C. natalensis*: Maxent = 0.87 ± 0.04; RF = 0.83 ± 0.05. Differently, AUCdiff values were consistently higher for the tuned RF models when compared to the Maxent ones, except for *C. darwini*, with mean ± sd values across the train-test CV splits corresponding to: (a) *C. brincki*: Maxent = 0.03 ± 0.04; RF = 0.21 ± 0.1; (b) *C. danielssoni*: Maxent = 0.02 ± 0.02; RF = 0.11 ± 0.12; (c) *C. darwini*: Maxent = 0.06 ± 0.04; RF = 0.06 ± 0.09; (d) *C. gahani*: Maxent = 0.06 ± 0.08; RF = 0.17 ± 0.16; (e) *C. natalensis*: Maxent = 0.02 ± 0.03; RF = 0.14 ± 0.07.

Both Maxent and RF managed to correctly predict medium-to-high habitat suitability (hereafter HS) in correspondence with the occurrence areas of all the target species (Figures 4 and 5). On the other hand, while in the obtained Maxent models medium-to-high predicted HS was mainly concentrated around the known occurrence localities, RF models extended predictions of medium HS values (i.e., HS = 0.4–0.5) far away from the species' presence points in some cases (e.g., *C. brincki* and *C. danielssoni*, Figure 4a,b; *C. natalensis*, Figure 5b). This resulted in RF overpredicting suitability when compared to Maxent for most of the target species outside the corresponding estimated EOOs (Supplementary Materials Figure S3). Thus, the higher AUCtest and lower AUCdiff values attained by Maxent may be primarily due to its lower predicted HS in correspondence with the background points comprising the test folds, which resulted in higher specificity (although specificity cannot be properly estimated with presence–background data [74]).

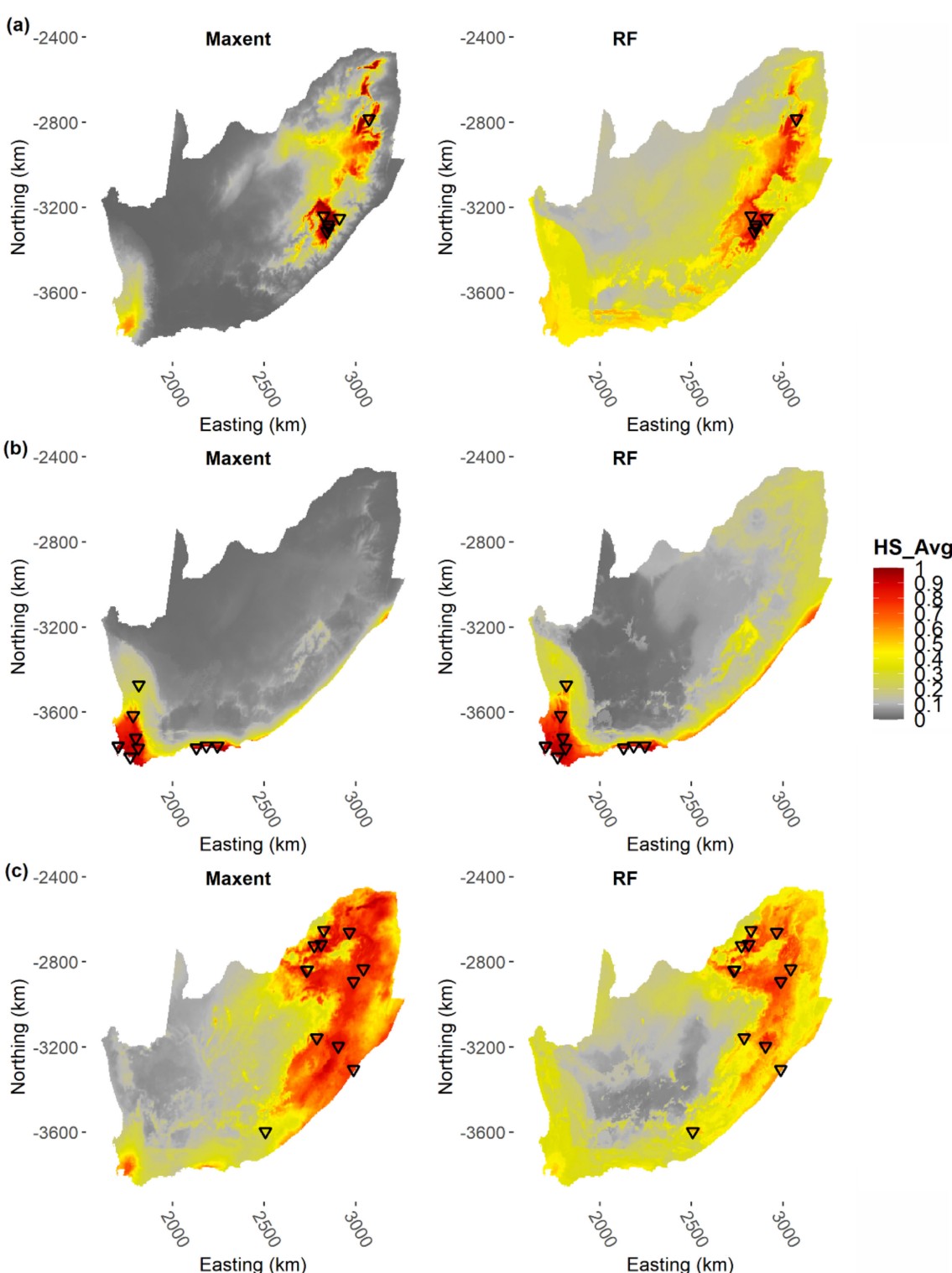

**Figure 4.** Weighted average predicted habitat suitability (HS_Avg) across the study region resulting from the tuned Maxent and Random Forests (RF) models for: (**a**) *Chaetocnema brincki*; (**b**) *C. danielssoni*; (**c**) *C. gahani*. Reversed black-contoured triangles indicate the species' occurrence localities.

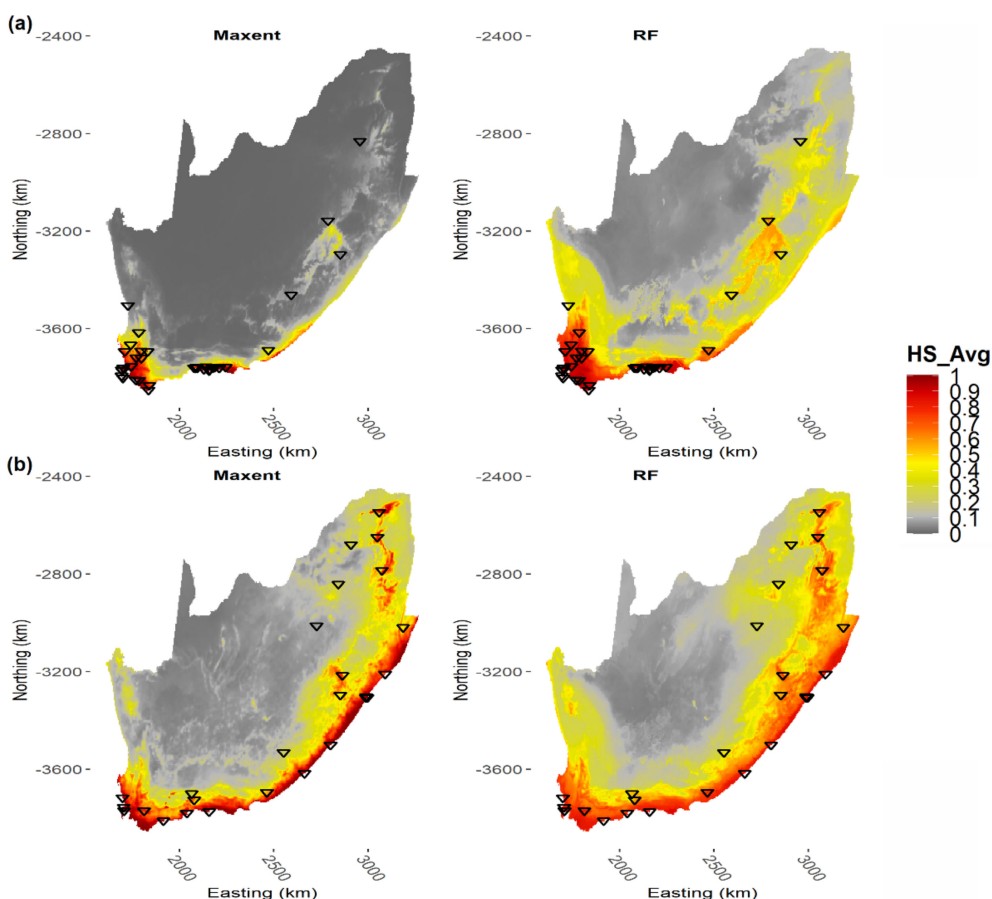

**Figure 5.** Weighted average predicted habitat suitability (HS_Avg) across the study region resulting from the tuned Maxent and Random Forests (RF) models for: (**a**) *Chaetocnema darwini*; (**b**) *C. natalensis*. Reversed black-contoured triangles indicate the species' occurrence localities.

For both Maxent and RF, Wgt_StdDev in predicted HS among the train-test CV splits was low across most of the study region for all the target species (Supplementary Materials Figures S4 and S5), indicating a high level of agreement between the tuned ENMs fitted on the different calibration datasets.

Despite the above-mentioned differences between Maxent and RF in the geographical arrangement of Wgt_Avg HS, the two algorithms mostly agreed in identifying the variables with the highest Std_Imp scores. Indeed, focusing on the top three variables in terms of Wgt_Avg Std_Imp scores, Maxent and RF shared at least two of them for all the target species (Supplementary Materials Table S3). Specifically, the tuned Maxent models selected for *C. brincki* related climatic suitability primarily to precipitation patterns (bio13 and bio15) and temperature of the wettest quarter (bio8), while in the corresponding RF models bio2 replaced bio15 among the most influential variables. Looking at the partial response curves obtained for these variables from the Maxent and RF models (Supplementary Materials Figure S6), the two algorithms agreed in predicting a negative relationship between HS and bio8. Differently, Maxent models predicted a sigmoid-like response to increasing precipitation of the wettest month (bio13) while RF predicted bimodal responses to this variable, though with higher variability among the LOO splits when compared to Maxent. For bio15, Maxent models from the different LOO splits agreed in predicting an increasing sigmoid-like response similar to that of bio13, while in RF models bio2 showed a weak negative relationship with HS. For *C. danielssoni*, increasing suitability with higher precipitation of the coldest month (bio19) emerged from both the Maxent and the RF models (Supplementary Materials Figure S7), with low variability among the LOO splits

and very high Wgt_Avg Std_Imp scores for this variable, especially in Maxent models (Supplementary Materials Table S3). Similarly to what emerged for *C. brincki*, predicted suitability for *C. danielssoni* is negatively related to bio8 (Supplementary Materials Figure S7), the variable with the second highest Wgt_Avg Std_Imp score for both Maxent and RF (Supplementary Materials Table S3). For *C. darwini*, negative responses to increasing values in bio2 and bio8 emerged from both Maxent and RF, although with noticeable variability between the two Checkerboard CV splits (Supplementary Materials Table S3, Figure S8); moreover, a bell-shaped response to bio19 peaking between 75 and 150 mm, depending on the considered Checkerboard CV split, emerged for this species from RF. Maxent and RF models shared all the top three variables for *C. gahani*, with particularly high Wgt_Avg Std_Imp scores for bio13 and bio2, followed by bio15 (Supplementary Materials Table S3): Maxent predicted monotonically increasing HS with higher values of both bio13 and bio15, and decreasing suitability at increasing bio2 values but with higher variability among the LOO splits (Supplementary Materials Figure S9); RF models also predicted a negative relationship between bio2 and predicted HS, while bell-shaped curves resulted for bio13 and bio15. Finally, for *C. natalensis* bio2 is by far the most influential variable for both Maxent and RF (Supplementary Materials Table S3), with sigmoid-like negative response curves resulting from both the Checkerboard CV splits (Supplementary Materials Figure S10); moreover, bell-shaped response to bio19, the second most influential variable, emerged from RF models with a peak around 150 mm, similarly to what resulted for *C. darwini*.

The species-specific 10th percentile training thresholds computed from the Wgt_Avg predictions obtained from the tuned Maxent and RF models were as follows: for *C. brincki*, Maxent = 0.52 and RF = 0.78; for *C. danielssoni*, Maxent = 0.60 and RF = 0.79; for *C. darwini*, Maxent = 0.12 and RF = 0.57; for *C. gahani*, Maxent = 0.7 and RF = 0.65; for *C. natalensis*, Maxent = 0.25 and RF = 0.51. It appears that ENMs fitted through the Checkerboard CV approach (i.e., for *C. darwini* and *C. natalensis*) had noticeably lower 10th percentile training thresholds than those fitted through LOO CV, especially for Maxent, suggesting that in the former case the ENMs predicted low HS for some of the presence points (see Figure 5).

For *C. brincki*, *C. danielssoni* and *C. gahani*, the binarized suitable pixels broadly corresponded to their actual occurrence areas, for both Maxent (Figure 6) and RF (Supplementary Materials Figure S11) and regardless of the used binarization threshold. For the remaining two species, the 10th percentile training threshold appeared as being more suited to properly predict climatic suitability across their EOOs, while using Wgt_Avg suitability $\geq 0.8$ as threshold made the binarized climatic suitability noticeably underestimate the distribution of *C. darwini* and *C. natalensis* in inland portions of their EOOs (see Figure 7 for Maxent, Supplementary Materials Figure S12 for RF).

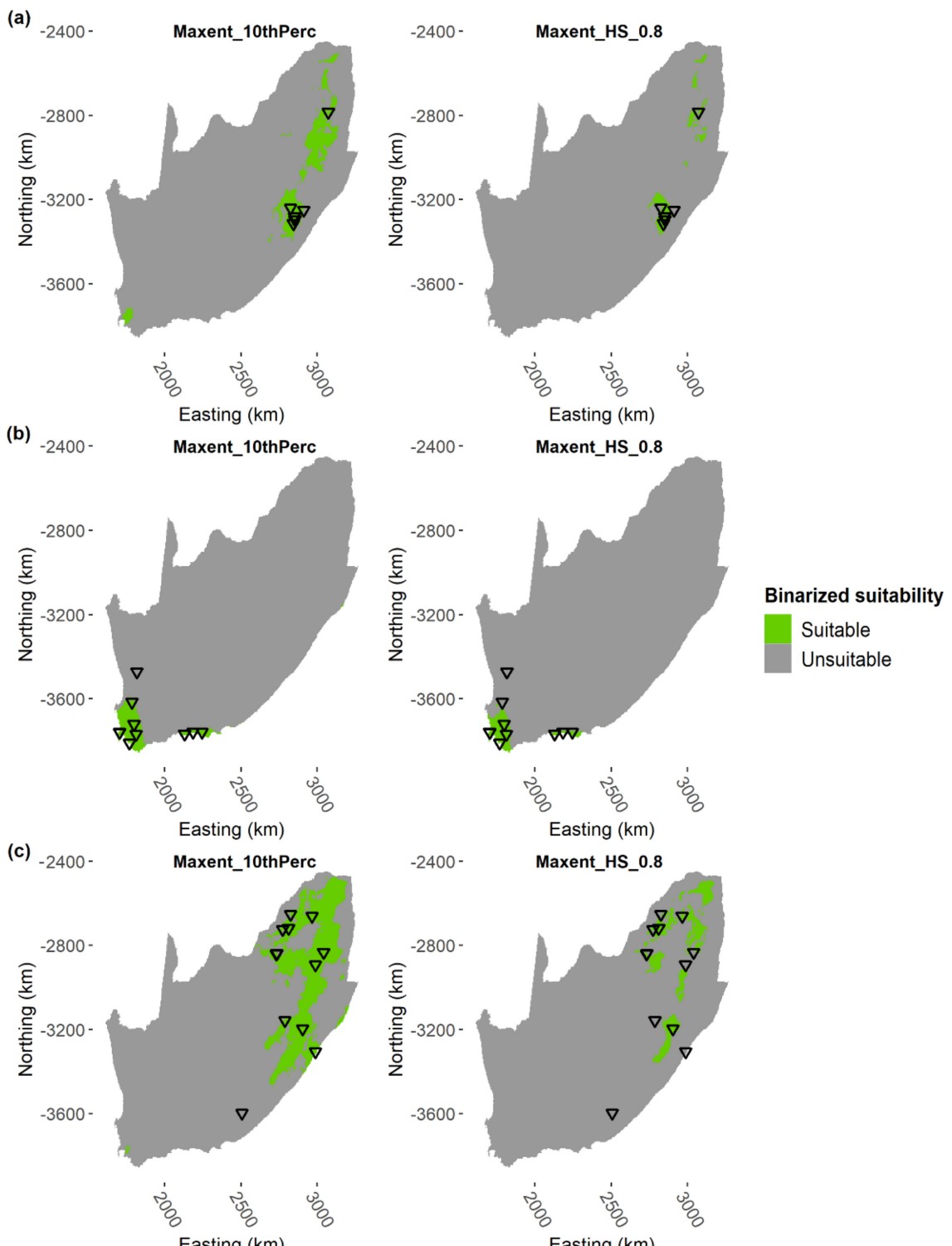

**Figure 6.** Binarized suitability resulting from the conversion of the Maxent-derived weighted average habitat suitability (Wgt_Avg HS) to binary predictions based on the 10th percentile training threshold (left maps) or on a threshold corresponding to Wgt_Avg HS ≥ 0.8 (right maps), for: (**a**) *Chaetocnema brincki*; (**b**) *C. danielssoni*; (**c**) *C. gahani*. Reversed black-contoured triangles indicate the species' occurrence localities.

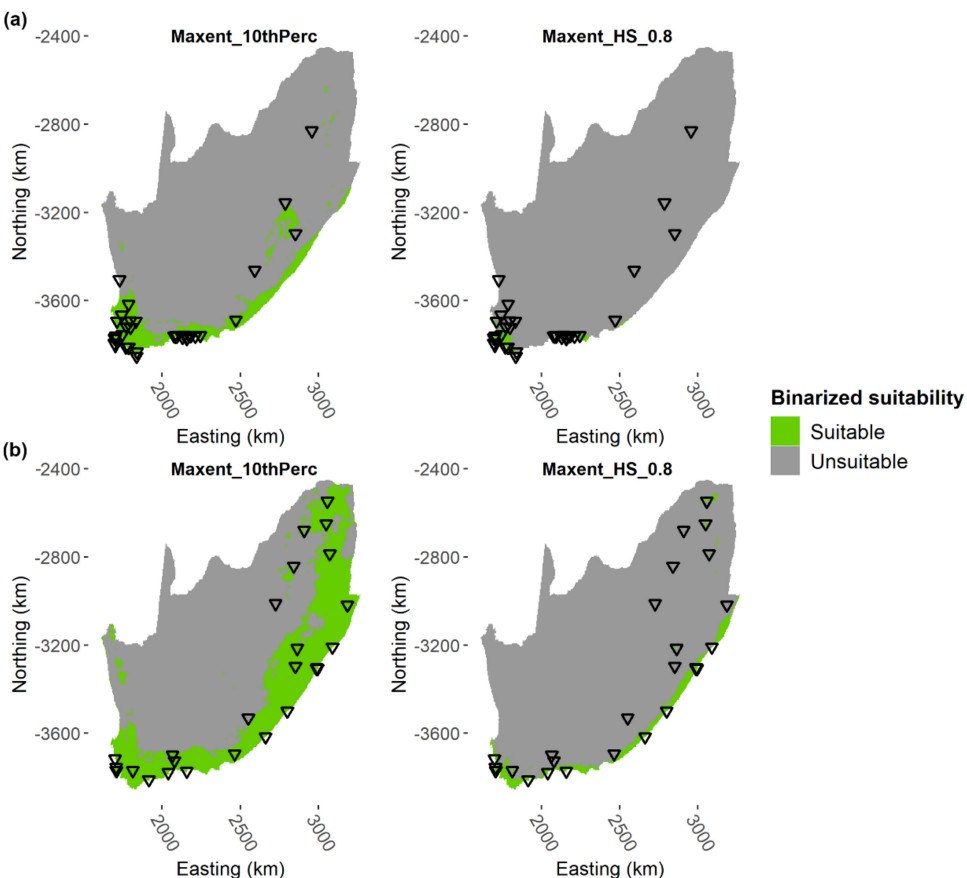

**Figure 7.** Binarized suitability resulting from the conversion of the Maxent-derived weighted average habitat suitability (Wgt_Avg HS) to binary predictions based on 10th percentile training threshold (left maps) or on a threshold corresponding to Wgt_Avg HS $\geq$ 0.8 (right maps), for: (**a**) *Chaetocnema darwini*; (**b**) *C. natalensis*. Reversed black-contoured triangles indicate the species' occurrence localities.

Climatically suitable areas for *C. brincki* showed high percentage cover (median cover ranging between 65% and 75%) by Temperate Grassland, Meadow and Shrubland (2.B.2), followed by Warm Temperate Forest (1.B.1), whose median coverage ranged from 10% to 30% depending on the considered algorithm, with higher median coverage for RF (Figure 8a). Temperate Grassland, and Meadow and Shrubland showed the highest percentage coverage (median ~75%) also within climatically suitable pixels individuated for *C. danielssoni* in all the algorithm × threshold combinations (Figure 8b), followed by Tropical Flooded and Swamp Forest (1.B.3), though with noticeably lower median percentage cover (between 5% and 10%). The highest median cover values for *C. gahani* (~85%) emerged for the Tropical Lowland Grassland, and Savanna and Shrubland (2.A.1) formations when RF-derived Wgt_Avg predictions were binarized using HS $\geq$ 0.8 as a threshold (Figure 8c); in this case, however, only the occurrences located in the north-eastern portion of the species' range fell within pixels predicted as suitable (see Supplementary Materials Figure S11), so this climate–vegetation association should being interpreted as limited to this subset of occurrences. Differently, when Wgt_Avg predictions for *C. gahani* were binarized using the 10th percentile training threshold the highest percent cover of climatically suitable areas emerged for Temperate Grassland, and Meadow and Shrubland (2.B.2), with a median value of about 35% for Maxent-derived predictions and 55% for RF-derived ones. Similarly to what emerged for *C. brincki*, Warm Temperate Forest (1.B.1) showed a not negligible percentage cover of climatically suitable pixels also for *C. gahani*, though with far lower median values (around ~5%) than for Temperate Grassland, and Meadow and Shrubland (2.B.2).

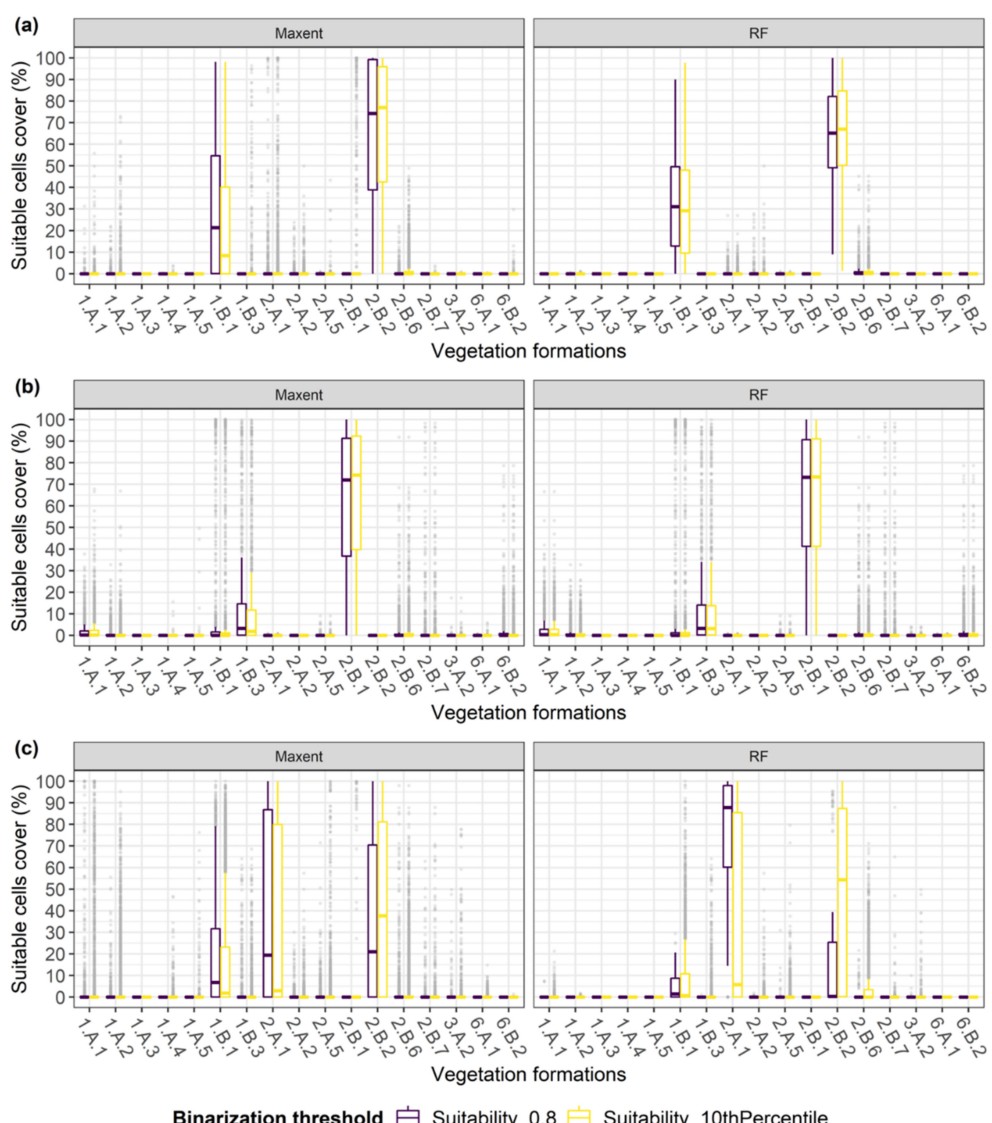

**Figure 8.** Percentage coverage by each of the 17 vegetation formations (see Table 1) of the pixels predicted as climatically suitable after binarizing the Maxent-derived (left) and RF-derived (right) weighted average predictions through either the 10th percentile training threshold or the HS $\geq$ 0.8 threshold, for: (**a**) *Chaetocnema brincki*; (**b**) *C. danielssoni*; (**c**) *C. gahani*.

Mediterranean Scrub and Grassland (2.B.1) was the predominant vegetation formation within climatically suitable areas individuated for *C. darwini*, with a median percentage cover ranging from 25% for the Maxent × 10th percentile combination to 70% for the RF × HS $\geq$ 0.8 one (Figure 9a); considering the 10th percentile training threshold, Tropical Seasonally Dry Forest (1.A.1) emerged as the formation with the second highest percentage cover of suitable pixels for this species, particularly in Maxent-derived binarized predictions. For *C. natalensis*, the percentage cover of climatically suitable pixels by the different vegetation formations varied noticeably with the chosen binarization threshold (Figure 9b): suitable areas individuated through the 10th percentile training threshold appeared to be primarily covered by Temperate Grassland, and Meadow and Shrubland (2.B.2), followed by Warm Temperate Forest (1.B.1), and Tropical Lowland Grassland, and Savanna and Shrubland (2.A.1); differently, pixels predicted as being suitable using the HS $\geq$ 0.8 threshold, primarily confined to the southern coastal region, were mainly covered by Tropical Seasonally Dry Forest (1.A.1) and Tropical Lowland Humid Forest (1.A.2) with median coverage values around 10%.

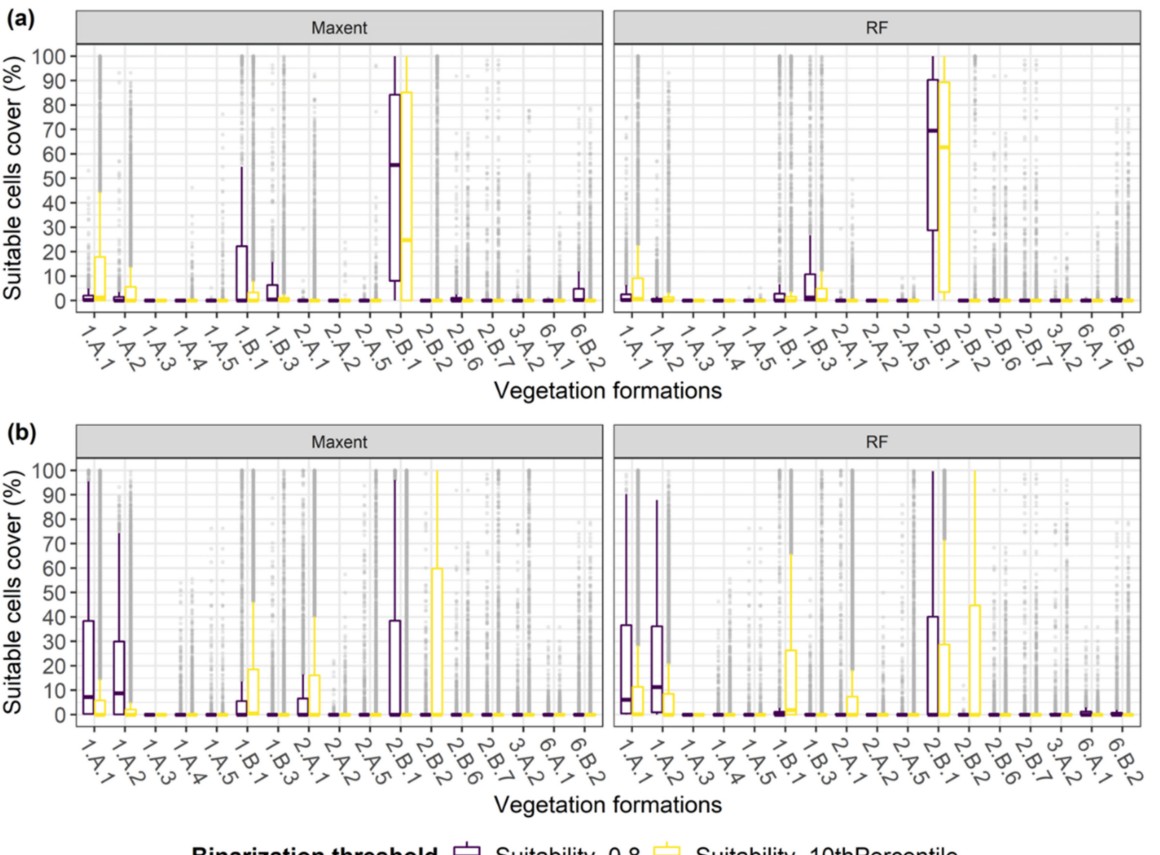

**Figure 9.** Percentage coverage by each of the 17 vegetation formations (see Table 1) of the pixels predicted as climatically suitable after binarizing the Maxent-derived (left) and RF-derived (right) weighted average predictions through either the 10th percentile training threshold or the HS $\geq$ 0.8 threshold, for: (**a**) *Chaetocnema darwini*; (**b**) *C. natalensis*.

The first three axes resulting from the PCA-Env cumulatively explained 55.8% of the environmental heterogeneity, in terms of bioclimatic conditions and vegetation cover, characterising the study region (PrinComp1: 25.3%; PrinComp2: 15.9%; PrinComp3: 14.6%). Considering the 2D environmental space defined by PrinComp1 and PrinComp2, the bioclimatic variables (particularly bio2 for PrinComp1 and bio9 for PrinComp2) contributed more than the percentage cover of the vegetation formations in defining these axes (Supplementary Materials Figure S13), although a noticeable contribution to PrinComp2 was provided by Temperate Grassland, and Meadow and Shrubland (2.B.2). In this 2D environmental space, niche densities of all the five *Chaetocnema* species overlapped to some extent (Figure 10), although density centres were segregated for all the species' pairs except for *C. danielssoni* and *C. darwini* which indeed attained the highest niche overlap value (*D* = 0.7), followed by *C. natalensis*-*C. darwini* (*D* = 0.36) and *C. natalensis*-*C. danielssoni* (*D* = 0.31). Interestingly, *C. brincki* showed very low overlap with *C. danielssoni* as well as with the two wide-ranging species *C. darwini* and *C. natalensis* (Table 3). Nonetheless, none of the species' pairs showed significant (at $p \leq 0.05$) niche divergence within the niche similarity tests. Instead, niche conservatism emerged for the pair *C. darwini*–*C. danielssoni* ($p = 0.01$), and for the pair *C. natalensis*-*C. danielssoni* ($p = 0.059$) if the threshold for type I error is relaxed up to $p \leq 0.1$.

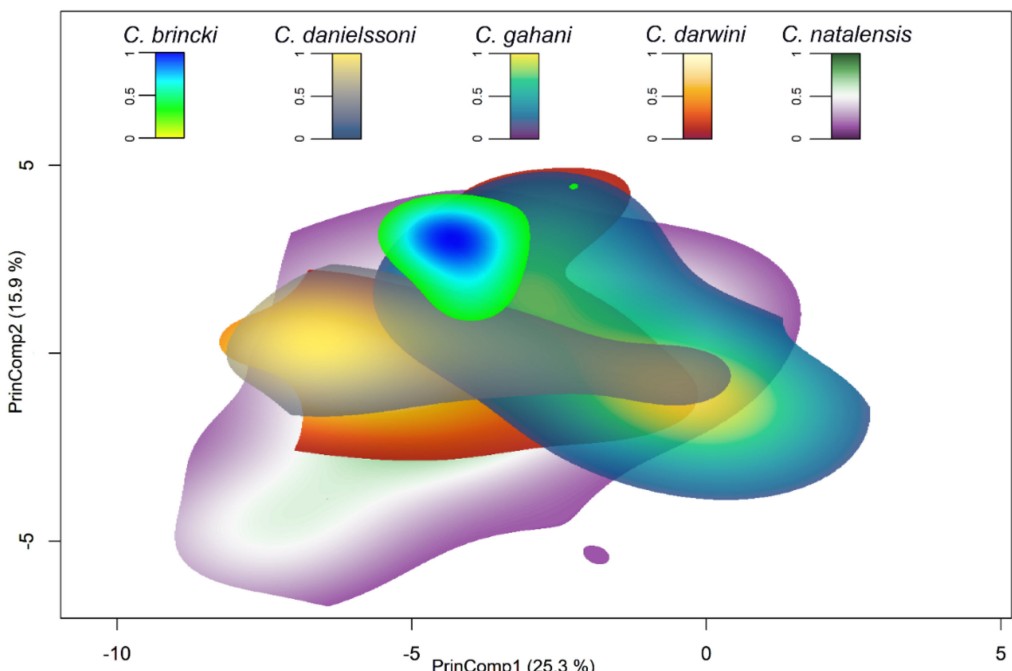

**Figure 10.** Kernel-based niche densities of the five target *Chaetocnema* species within the 2D environmental space encompassed by the top two principal components (in terms of percentage explained variance, values in brackets within axes labels) resulting from a PCA summarizing environmental heterogeneity in the study region based on the 19 bioclimatic variables and on the percentage coverage from the 17 vegetation formations.

**Table 3.** For each *Chaetocnema* species' pair reported: overlap in niche density as measured through the Schoener's *D* index; *p*-value of the niche similarity test conducted testing for niche divergence as alternative hypothesis; *p*-value of the niche similarity test conducted testing for niche conservatism as alternative hypothesis.

| Species' Pair | Schoener's *D* | SimTest_Divergence | SimTest_Conservatism |
|---|---|---|---|
| *C. darwini–C. danielssoni* | 0.7 | 1 | 0.01 |
| *C. natalensis–C. darwini* | 0.36 | 0.822 | 0.149 |
| *C. natalensis–C. danielssoni* | 0.31 | 0.98 | 0.059 |
| *C. natalensis–C. gahani* | 0.2 | 0.624 | 0.465 |
| *C. gahani–C. darwini* | 0.14 | 0.604 | 0.327 |
| *C. gahani–C. brincki* | 0.12 | 0.911 | 0.228 |
| *C. gahani–C. danielssoni* | 0.12 | 0.683 | 0.257 |
| *C. natalensis–C. brincki* | 0.04 | 0.693 | 0.317 |
| *C. darwini–C. brincki* | 0.02 | 0.703 | 0.287 |
| *C. danielssoni–C. brincki* | 0.01 | 0.723 | 0.307 |

## 4. Discussion

The southernmost African region encompassed by the Republic of South Africa, Lesotho and Swaziland hosts very diversified vegetation formations, ranging from desert or semi-desert grassland and shrubland in the north-west, to temperate grassland, meadow and shrubland, interspersed by patches of Afromontane forest, characterizing the Central Plateau and the highlands in the north-east, to the southern coastal humid or seasonally dry forests [24,75]. This diversity derives from paleoclimatic and orographic processes, and is currently linked to three main air circulation cells: an anticyclonic system from the Indian Ocean, bringing summer rainfalls in the eastern portion of the region; a westerly Polar Front system from the South Atlantic Ocean, producing particularly rainy winters followed by dry summers in the south-west; the cool and dry trade winds from

South Atlantic Ocean, bringing aridity in the north-west [26]. The variety of vegetation formations and climatic conditions in turn have produced a noticeable richness in beetles from various families, such as Curculionidae, Chrysomelidae and Scarabeidae [24,26–28], with several species being rare [28].

Within Chrysomelideae, the *Chaetocnema* genus is very diversified in the Afrotropics and the five target species investigated here represent South African endemics or sub-endemics (*C. gahani* and *C. natalensis*). Given the paucity of occurrence records available for some of these species (e.g., *C. brincki*, *C. danielssoni*) and their geographically clustered arrangement, we differentiated the model fitting and evaluation procedures using either LOO cross-validation or checkerboard spatial blocking, depending on the sample size of the considered species, to increase the spatial independence of the test data. This stratagem, along with the tuning of Maxent and RF parameters to select the ones best suited to the specific dataset, permitted us to obtain ENMs with a noticeably high discrimination performance for most species and without relevant overfitting on training data. In this manner, the AUC-based weighted average predictions obtained for both algorithms are likely to not suffer from strong biases deriving from spatial autocorrelation or poor model tuning [44,54,57]. Moreover, it is noteworthy that the tuning process led to different optimal parameterizations, for both Maxent and RF, depending on the considered species: this reinforces previous claims about the need to avoid automatically relying on default parameterization for these machine learning algorithms, and instead explore various parameter settings to individuate the one(s) best fitting the specific modeling task (e.g., extrapolation versus interpolation) and the characteristics of the available datasets [44,54].

The weighted average predicted suitability, particularly from the tuned Maxent models, suggests that the available occurrence records of the target *Chaetocnema* species cover most of the climatically suitable areas, with the exception of *C. brincki* and *C. gahani* for which some highly suitable areas in the north-east (Limpopo Province for both species, KwaZulu-Natal for *C. gahani*) have no observed records so far. A drop in predicted suitability with a mean temperature of the wettest quarter (i.e., bio8) higher than 15 °C is shared by *C. brincki* and *C. danielssoni*: for the former, this relates to a relatively low temperature in the eastern highlands during summer; for *C. danielssoni* this pattern may instead be linked to the southern Polar Front determining a moist winter (i.e., low temperatures coupled with intensive precipitation) in the lowlands of the Western Cape Province. Nevertheless, while climatic suitability for *C. danielssoni* is also predicted to be strongly related to heavy rainfall during the coldest quarter (i.e., bio19), coherently with its EOO restricted to south-western South Africa, suitability for *C. brincki* is predicted to mainly depend upon precipitation higher than 100 mm during the wettest (summer) month (i.e., bio13). Similar to *C. brincki*, climatic suitability for *C. gahani* steadily increases with bio13 values higher than 100 mm; moreover, suitability for both species is strongly related to marked precipitation seasonality, here again reflecting their distribution that is limited to the eastern portion of Southern Africa. Nevertheless, climatic preferences of these two species differ in that suitability for *C. gahani* is negatively related to diurnal thermal excursion (i.e., bio2) rather than being limited by the average temperature of the wettest quarter, possibly explaining its wider EOOs. The wide EOO of *C. darwini*, ranging from the south-western lowlands to the mountainous regions of Lesotho and north-eastern South Africa, makes the climate-occurrence patterns modeled for this species a mix of those characterizing the three previously analyzed species: indeed, both Maxent and RF predicted decreasing suitability with increasing values of bio2 and bio8, one of the two tuned Maxent models predicted a steep increase in suitability with bio13 > 100 mm, and both the tuned RF models predicted a bell-shaped response to bio19. Finally, modeled suitability for *C. natalensis* partly differs from those of the other species because it is predicted, particularly by Maxent, to be heavily reduced as diurnal temperature range exceeds 7–8 °C; moreover, RF models also related suitability for *C. natalensis* with bio13 and bio19 through bell-shaped curves with the optimum placed around 100 mm and 150 mm, respectively. These modeled climate-occurrence patterns are coherent with *C. natalensis* being totally absent from the arid Northern Cape Province.

Temperate Grassland, and Meadow and Shrubland were the dominant vegetation formations, in terms of percentage cover of climatically suitable areas, for *C. brincki*, *C. danielssoni* and *C. gahani*, confirming previous observations [24]. For *C. danielssoni*, the flooded forests of the Western Cape Province also represent an important component of climatically suitable areas. *Chaetocnema brincki* and *C. gahani* also share the noticeable coverage of their climatically suitable areas by the Warm Temperate Forest formation. The southern Mediterranean Scrub and Grassland represents instead the vegetation formation primarily associated with areas predicted as climatically suitable for *C. darwini*, although also the Tropical Seasonally Dry Forest and the Temperate Flooded and Swamp Forest formations, this latter likely limited to the south-western portion of the species' range, attained moderate coverage values. *Chaetocnema natalensis*, coherently with its range spanning almost the entire west-east gradient of the study region, showed climatic suitability associated to various vegetation formations, ranging from Temperate Grassland, Meadow & Shrubland, to Lowland Grassland, and Savanna and Shrubland, to Tropical Lowland Humid Forest.

The niche overlap analysis performed through the PCA-Env approach confirmed the similarities and differences emerging from the ENMs among the five target *Chaetocnema* species, and provided a novel, important insight: *Chaetocnema danielssoni* significantly shares the portion of its environmental niche represented by the considered predictors with *C. natalensis* and, more pronouncedly, with *C. darwini*, despite being evidently more narrow-ranging than these latter species. This niche conservatism pattern suggests that the current occurrence localities of *C. danielssoni* are limited to south-westernmost South Africa for reasons not relating to climate or localized vegetation formations. Moreover, given that strict trophic association with one or few plant species is quite rare among flea beetles [76], it is improbable that the restricted distribution of *C. danielssoni* is due to its inability to feed outside its current range. The factors which have constrained *C. danielssoni* within the Western Cape Province could be further investigated in the future by applying molecular techniques to shed light on the historical demography of this species and its neighbouring ones.

## 5. Conclusions

The implemented ecological niche modelling framework permitted the deepening of the current understanding of the climatic drivers shaping the distribution patterns of South African *Chaetocnema* species, also highlighting the associations between suitable climatic conditions for these flea beetles and the diversified vegetation formations of the study region. Moreover, the obtained maps of climatic suitability suggest that, at least for some of the considered species, additional sampling campaigns in eastern South Africa may provide new occurrence records, contributing to reduction in the Wallacean shortfall for these (and likely other ecologically similar) phytophagous beetles. Both the algorithms here implemented, Maxent and Random Forests, showed noticeable explanatory power coupled with good generalizability (low overfitting); thus, they were confirmed to be a fundamental part of the ecological niche modelling toolbox, so long as they are properly tuned.

**Supplementary Materials:** The following supporting information can be downloaded at https://www.mdpi.com/article/10.3390/d14020100/s1, Table S1: Target species' occurrence coordinates; Table S2: Parameterization and discrimination performance of tuned models; Table S3: Variables importance scores; Figure S1: Correlograms from preliminary models; Figure S2: Checkerboard cross-validation blocks; Figure S3: Between-algorithm suitability difference; Figure S4: Suitability standard deviation for *Chaetocnema brincki*, *C. danielssoni* and *C. gahani*; Figure S5: Suitability standard deviation for *Chaetocnema darwini* and *C. natalensis*; Figure S6: Response curves for *Chaetocnema brincki*; Figure S7: Response curves for *Chaetocnema danielssoni*; Figure S8: Response curves for *Chaetocnema darwini*; Figure S9: Response curves for *Chaetocnema gahani*; Figure S10: Response curves for *Chaetocnema natalensis*; Figure S11: Binarized suitability for *Chaetocnema brincki*, *C. danielssoni* and *C. gahani*; Figure S12: Binarized suitability for *Chaetocnema darwini* and *C. natalensis*; File S1: the whole R script written

to perform all the analyses presented in this article; a CSV file containing coordinates (in WGS84 reference system) of occurrence records of the target *Chaetocnema* species.

**Author Contributions:** Conceptualization, M.B. and F.C.; methodology, F.C. and M.B.; data curation, M.B. and P.D.; software, F.C.; formal analysis, F.C.; investigation, F.C., P.D. and M.B.; resources, M.B.; writing—original draft preparation, F.C.; writing—review and editing, F.C., P.D. and M.B. All authors have read and agreed to the published version of the manuscript.

**Funding:** This research received no external funding.

**Institutional Review Board Statement:** Not applicable.

**Informed Consent Statement:** Not applicable.

**Data Availability Statement:** Coordinates of occurrence points used to fit the ENMs and perform the niche overlap analysis for the target *Chaetocnema* species are provided in Supplementary Materials Table S1, as well as in a separate .csv file; a commented version of the R script written to conduct the analyses is also provided as Supplementary Materials File S1.

**Acknowledgments:** We thank two anonymous reviewers for their stimulating comments on a previous version of this article.

**Conflicts of Interest:** The authors declare no conflict of interest.

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
