# Peer review of "Fine-Tuned Ecological Niche Models Unveil Climatic Suitability and Association with Vegetation Groups for Selected Chaetocnema Species in South Africa (Coleoptera: Chrysomelidae)"

_diversity, doi:10.3390/d14020100_

Round 1

Reviewer 1 Report

The study is well designed and presented.

But Chaetocnema, as most of leaf beetles, are specialized phytophages. In many cases their areas depend of climatic conditions and vegetation formations indirectly through their host plant distribution. It is clear that in many cases no information is available, but as for me, any research on leaf beetle ecology should include discussion about their host specialization. At least in the Introduction, if nothing is known about it. 

For example, it is quite probable hypothesis that the limitation of the distribution of Ch. danielssoni is associated with the limitation of the distribution of the host plant, but this assumption has not even been made.

Author Response

Reviewer 1

The study is well designed and presented.

Dear Reviewer 1, thank you for having appreciated our efforts. Please note that line numbering cited in the following comments refers to the “clean” version of the revised manuscript.

But Chaetocnema, as most of leaf beetles, are specialized phytophages. In many cases their areas depend of climatic conditions and vegetation formations indirectly through their host plant distribution. It is clear that in many cases no information is available, but as for me, any research on leaf beetle ecology should include discussion about their host specialization. At least in the Introduction, if nothing is known about it.

Thank you for this suggestion; we added a sentence in the Introduction (see L. 71-74) indicating the main plant families to which Chaetocnema species are usually associated.

For example, it is quite probable hypothesis that the limitation of the distribution of Ch. danielssoni is associated with the limitation of the distribution of the host plant, but this assumption has not even been made.

Thank you for this suggestion. However, given that very strict trophic association with one or few plant species is rare among flea beetles (Jolivet & Hawkeswood, 1995), we think it is improbable that the current restricted distribution of C. danielssoni is due to its inability to find suitable trophic resources outside its current range. Nonetheless, we added a sentence at the end of the Discussion (see L. 614-617) to cite this hypothesis.

Jolivet P., Hawkeswood T.J. Host-Plants of Chrysomelidae of the World. An Essay about the Relationships between theLeaf-Beetles and their Food-Plants. Backhuys, Leiden, Netherlands 1995.

Reviewer 2 Report

General comments

The manuscript (ms) submitted as an “article” to the journal Diversity. The scopes and the suggested outline of the ms suits well with the journal’s mission and I’m sure that the ms will attract many readers to the journal. The topic has a high novelty with a strong methodological advances in SEM models. The ms very well-written the language and style are clear and concise. Some really minor structural improvement might be addressed in results. I slightly missed a methodological tone more emphasized in the text in terms of the model parametrization and performances’ projection for future use.

I think that the authors made a neat ms which is ready for publication after a minor revision.

Specific comments

Line 26-27 - I may suggest to close the abstract with a sentence about the major implication of RF and MaxEnt models in species conservation actions.

Line 114 – I have just wondering whether it would be a good idea to integrate the table intro the legend of the figure 1.

Line 189 - The section between lines 193 -218 describes well the content of the table 2 , thus I may suggest to integrate it into the text, or keep the table in the ESM.

Line 304 – I may suggest the following outline for the results section: 1) comparison of the model performances and parametrization globally, focus on the overall performance and predictability power; 2) report the species level responses for each RF and MaxEnt and and then the 3.) niche overlap models.

Line 344 – Figure 2 – I think the use of the same symbols and colors for each species as in fig1, may improve the coherency between figure especially for figs 3-6.

Line 351 - Table 3 should be placed into ESM.

Line 357 – The species symbols are not so visible on Fig 3 and 4.

Line 423- Table 4 should be placed into ESM.

Line 429 – The species symbols are not so visible on Fig 5 and 6.

Line 506 – The table should be placed into ESM

Line 620 - I would be flattered to include the R script of these models, or at least complete codes for one species and a working example for readers.

Author Response

Reviewer 2

The manuscript (ms) submitted as an “article” to the journal Diversity. The scopes and the suggested outline of the ms suits well with the journal’s mission and I’m sure that the ms will attract many readers to the journal. The topic has a high novelty with a strong methodological advances in SEM models. The ms very well-written the language and style are clear and concise. Some really minor structural improvement might be addressed in results. I slightly missed a methodological tone more emphasized in the text in terms of the model parametrization and performances’ projection for future use.

I think that the authors made a neat ms which is ready for publication after a minor revision.

Dear Reviewer 2, thank you for having appreciated our efforts. In this revised version we tried to highlight more in detail the methodological implications (in terms of Maxent and Random Forest tuning) of our modelling workflow, and how these latter may be useful to researchers wishing to implement these algorithms in ENMs studies. Please note that line numbering cited in the following comments refers to the “clean” version of the revised manuscript.

 Specific comments

Line 26-27 - I may suggest to close the abstract with a sentence about the major implication of RF and MaxEnt models in species conservation actions.

Thank you for this suggestion; we added a final sentence of this kind in the Abstract.

Line 114 – I have just wondering whether it would be a good idea to integrate the table intro the legend of the figure 1.

Thank you for this suggestion; anyway, we prefer to maintain Figure 1 as it is because we already tried to include the names of the vegetation formations, instead of their alphanumerical codes, in the legend but in this manner the Figure became too messy. 

Line 189 - The section between lines 193 -218 describes well the content of the table 2 , thus I may suggest to integrate it into the text, or keep the table in the ESM.

Thank you for this suggestion but, given that the Methods section is already quite long and not so straightforward, we preferred to maintain the different tested parameterizations in Table 2 to let the reader easily individuate them.

Line 304 – I may suggest the following outline for the results section: 1) comparison of the model performances and parametrization globally, focus on the overall performance and predictability power; 2) report the species level responses for each RF and MaxEnt and and then the 3.) niche overlap models.

Thank you for this suggestion. We indeed ameliorated the first section you cite by briefly discussing inter-species differences in parameterization and discrimination performance (AUCtest and AUCdiff) for both Maxent and Random Forests (see L. 325-349).

In our opinion, points 2) and 3) were already comprehensively described in the previous version so we didn’t change the corresponding paragraphs.

Line 344 – Figure 2 – I think the use of the same symbols and colors for each species as in fig1, may improve the coherency between figure especially for figs 3-6.

Thank you for the suggestion; anyway, the symbols and shapes for species’ presence localities in Figure 1 were specifically selected to let them be visible once overlaid on the colours depicting the vegetation formations, and the map in Figure 1 was produced in the Qgis software while those in Figs. 2-6 were directly extracted from the R script used to conduct all the analyses. Thus, we preferred to maintain the symbols of species’ occurrences in Fig. 1 distinct from those in the other Figures; nonetheless, we changed these latter (from blue-contoured triangles to black-contoured reversed triangles) and increased their size throughout Figs. 2-6, so as to make them clearly visible and consistent across these Figures.

Line 351 - Table 3 should be placed into ESM.

Thank you for this suggestion; we moved this table to Supplementary Materials (it is now Table S2).

Line 357 – The species symbols are not so visible on Fig 3 and 4.

Please see the previous comment referring to Figure 2.

Line 423- Table 4 should be placed into ESM.

Thank you for this suggestion; we moved this table to Supplementary Materials (it is now Table S3).

Line 429 – The species symbols are not so visible on Fig 5 and 6.

Please see the previous comment referring to Figure 2.

Line 506 – The table should be placed into ESM

Thank you for this suggestion, but we preferred to maintain this Table in the main text so as to provide direct information to readers about the niche overlap patterns for all the species’ pairs.

Line 620 - I would be flattered to include the R script of these models, or at least complete codes for one species and a working example for readers.

Thank you for this suggestion. We included as Supplementary Materials a commented version of the complete R script used to conduct all the analyses (named “Chaetocnema_SouthAfrica_ENMs_FullCode.R”), and a .csv file (named ''Chaetocnema_SouthAfrica_OccurrWGS84.csv") with WGS84 coordinates of the occurrence localities for all the target species, so that readers could full replicate our workflow and possibly use it for their own research.